# The role of educational attainment and quality in U.S. regional variation in prevalence of dementia and CIND

Jennifer A. Ailshire[1]*, Mateo P. Farina[2], Heide Jackson[3], Katrina M. Walsemann[4]

1 Leonard Davis School of Gerontology, University of Southern California, Los Angeles, California, United States of America, 2 Department of Human Development and Family Sciences, University of Texas at Austin, Austin, Texas, United States of America, 3 Population Research Center, University of Maryland, College Park, Maryland, United States of America, 4 School of Public Policy, University of Maryland, College Park, Maryland, United States of America

* ailshire@usc.edu

## Abstract

There are striking disparities in dementia prevalence across regions of the U.S. Education is one of the most important risk factors for dementia. Level and quality of education varied geographically for older cohorts of U.S. adults, potentially contributing to regional differences in dementia prevalence. This study links historical state education quality data to respondents ages 65 and older in the Health and Retirement Study to determine the extent to which geographic disparities in dementia and cognitive impairment with no dementia (CIND) can be attributed to state-level measures of education quality. Older adults educated in states with better resourced education systems had lower prevalence of dementia (Relative risk ratio (RRR): 0.81; CI: 0.75, 0.87) and CIND (RRR: 0.89; CI: 0.84, 0.93), while those educated in states where more school funding came from state rather than local sources had higher prevalence of dementia (RRR: 1.12; CI: 1.01, 1.23) and CIND (RRR: 1.10; CI: 1.03, 1.16). Educational attainment does not explain the higher prevalence of dementia or CIND in the US South, but state-level education quality fully accounted for higher prevalence of dementia and CIND in the South. Finally, state-level education quality indicators were more strongly associated with dementia and CIND among those with less education These findings suggest education does in fact explain regional disparities in dementia and cognitive impairment, particularly between the South and other regions, but that it is the educational environment that matters more for geographic differences than educational attainment in these cohorts.

## Introduction

Dementia affects more than 7 million people in the United States [1], and places a substantial burden on families, communities, and healthcare systems [2,3].

**Data availability statement:** Data cannot be shared publicly because state of residence can be identified from our data set and that is considered identifying information and restricted data by the Health and Retirement Study. Data can be access with an approved restricted data application and access to the MiCDA virtual data enclave. The application process to access these data is described here: https://hrs.isr.umich.edu/data-products/restricted-data/vdi.

**Funding:** This work was funded by the National Institute on Aging (grants R01AG055481 and P30AG043073).

**Competing interests:** The authors have declared that no competing interests exist.

Geography has become a major fault line in the U.S. reflecting vast inequalities in health, including differential dementia risk by region [4–9]. For instance, one study of 2008 Medicare beneficiary data found significant state variation in dementia prevalence, ranging from about 6–7% in Midwestern and Western states to 9–10% in Southern states [5]. In addition, a recent study documented stark differences in dementia prevalence by Census division [4]; among a nationally representative sample of adults 65 and older in 2012, dementia prevalence was 5.4% in the West North Central and 13.8% in the East South Central – a difference of over 8 percentage points. To put this difference into perspective, a recent report on national dementia prevalence using 2016 data found a 4-percentage point difference between older adults with less than high school education (13%) and those with a college degree (9%). Thus, geographic disparities are significant, larger even than differences between education groups, and yet are not well understood [10].

Education is considered among the most important determinants of dementia risk [11]. Studies consistently show that individuals with greater educational attainment exhibit a reduced lifetime risk of dementia and experience a delayed onset of the disease [12]. The relationship between education and dementia may be explained in part by cognitive reserve theory, which posits that higher levels of education provide individuals with greater cognitive resilience against age-related brain changes [13,14]. According to recent studies, declines in dementia prevalence and incidence documented at the national level are likely explained by increasing levels of educational attainment across birth cohorts of older adults [15,16]. Additionally, research has also found that subnational geographic variation in dementia trends was largely explained by differences in educational attainment over time across U.S. regions [4] with education gradients in dementia particularly pronounced in the South [9].

While much of the existing research has focused on educational attainment as an explanation for geographic patterns in dementia, education quality, which varies by location, is also likely a key factor. Characteristics reflecting quality of education, such as school term length, per-pupil funding, and pupil-teacher ratio, may represent access to cognitively enriching opportunities during childhood that are essential for developing cognitive and brain reserve [17]. Prior research has shown that exposure to more advantaged educational systems – at the state and/or local level – is associated with better cognitive function among older adults, independent of educational attainment [18–24]. Moreover, large geographic differences in education quality existed when current cohorts of older adults were attending school [25], particularly in the U.S. South, which historically underinvested in their public schools [26]. Thus, education quality may be a key determinant of geographic disparities in dementia.

However, some evidence suggests educational quality is more consequential for the cognitive health of individuals with less educational attainment. For example, one study of older adults in Alabama found that low quality education was associated with poorer cognitive function, but only among adults with less than a high school education [18]. Education quality may be particularly important for individuals who spent less time in school and therefore had less exposure to complex cognitive tasks and

intellectually enriching activities. The South has historically had lower levels of educational attainment and poorer quality education systems [25], and the joint impact of these historical patterns might explain the higher risk of CIND and dementia observed in the South relative to other regions [4,5].

The present study aims to conduct a comprehensive analysis of the relationship between education and geographic variation in CIND and dementia among U.S. older adults. We expect that both fewer years of educational attainment and poor education quality will be associated with increased CIND and dementia risk, but that state-level educational quality will matter more for people with lower levels of education. We will determine the extent to which both individual educational attainment and state-level education quality, jointly and independently, explain regional disparities in CIND and dementia prevalence. We also hypothesize that because education systems in the South were historically under-resourced, differences in CIND and dementia between the South and other regions will be explained by differences in education quality. By examining the multifaceted nature of education in the context of geographic disparities in dementia, this study will advance our understanding of how education contributes to dementia disparities.

## Materials and methods

### Data

Individual-level data come from the Health and Retirement Study (HRS), a nationally representative, longitudinal study of US adults over age 50. Using a steady-state design, the HRS sample is replenished with younger cohorts about every 6 years. Since 1992, the HRS has conducted core interviews with age-eligible respondents and their spouses approximately every 2 years. The HRS is a multi-stage area probability sample of age-eligible households selected from primary sampling units chosen from U.S. Metropolitan Statistical Areas (MSAs) and non-MSA counties, with an oversampling of minorities and the oldest-old. This complex sampling design allows for nationally representative analysis of census regions to be performed at the population level.

State-level data on public education systems were drawn from the publicly available State-Education Contextual Data Resource (SECD-R), which compiles and standardizes historical information on school systems across U.S. states from 1920 to 1974 [27]. These data come from the Biennial Surveys of Education (1919/20–1957/58) and the Statistics of State School Systems (1959/60–1971/72). All data were reported biennially from 1919/20–1955/56 for 48 states, the District of Columbia. We excluded Alaska and Hawaii because they did not become U.S. states until 1959 and we don't have complete data for these states. We linked HRS respondents to state-level education data from the year when they were aged 10 based on respondent reports of the state where they lived most of the time they were in primary/secondary school or around age 10. About 23% of respondents in this sample (n = 2,028) resided in a different region in childhood than the region where they lived at the time of their interview.

We use publicly available data on state-level income inequality that includes estimates of income concentration within each state, reported as the share of total income held by the top 10% versus the bottom 90% of earners [28]. We linked HRS respondents to state-level income inequality data from the year when they were aged 7 based on respondent reports of the state where they lived most of the time they were in primary/secondary school or around age 10.

### Sample

Of the 42,572 HRS respondents included on the merged HRS interview file, we restricted our sample to the respondents born between 1913 and 1945 (n = 22,828), who were born in the United States (n = 20,676), resided in the continental United States while in school (n = 18,664), were last interviewed in the HRS after 2010 (n = 11,550), did not move between the 2010 and 2018 HRS interviews (n = 11,297), had a non-missing cognition measures at least one wave between 2010 and 2018 (n = 8,780), and had complete data on our model controls (n = 8,771). Our final sample size is 8,771 respondents contributing 33,253 observations (see S1 Figure in S1 Appendix).

## Measures

**Cognitive status.** HRS uses information from respondents, proxies, and interviewers to identify cognitive impairment and classify individuals as having no cognitive impairment, having cognitive impairment with no dementia (CIND), or as having dementia. Respondents were administered the Telephone Instrument for Cognitive Status (TICS) to assess cognitive function at each interview either by phone or face-to-face (and via web for a subset of the sample in 2018). The cognitive assessment consists of tests of immediate and delayed recall, serial 7s subtraction test, and counting backward. Scores from all items in the cognitive assessment are summed into a composite score of cognitive function that ranges from 0 to 27; higher scores reflect better cognitive function. We used cut-points defined by Langa-Weir, which classifies respondents with a score of 0–6 as having cognitive impairment consistent with dementia, 7–11 as having cognitive impairment but no dementia, and a score of 12 or above as normal cognitive functioning [29]. For respondents who completed the interview via web in 2018, we followed recent recommendations to consider a score of 12 as the upper bound for CIND, given evidence of mode effects on TICS scoring in web-based administration [30]. Cognitive status classification was also determined for respondents who had a proxy complete their interview and thus did not take the cognitive assessment. We used proxy assessment of the respondent's memory and proxy reports of respondent limitations in five instrumental activities of daily living (IADL), as well as the interviewer's assessment of the respondent's level of difficulty in completing the interview due to cognitive limitation. Scores on this assessment range from 0–11, with scores of 6–11 indicating cognitive impairment consistent with dementia, 3–5 indicating cognitive impairment without dementia (CIND), and 0–2 indicating normal cognitive function. This categorization shows good predictive ability when compared with classification from a consensus panel of experts in neuropsychiatric assessments of dementia [31]. We used the 3-category variable with respondents classified as having dementia, having cognitive impairment without dementia, and normal cognitive function.

**Current region of residence.** We determined region at the time of the respondents' interview from 2010–2018 using U.S. Census Bureau definitions of statistical regions (Northeast, South, Midwest, West). We excluded respondents who moved across U.S. regions between 2010 and 2018 (n = 253).

**State education quality.** We created summary scores of annual state-level education quality using seven-indicators: *pupil-teacher ratio* (average ratio of number of students in daily attendance to number of primary and secondary teachers), *teacher salary* (average salary of primary and secondary school teachers), *per-pupil spending* (total spending per pupil), *per-pupil revenue from local sources* (total local revenue per pupil), *per-pupil revenue from state sources* (total state revenue per pupil), *percent of revenue from local sources,* and *percent of revenue from state sources.* These indicators were selected based on their conceptual relevance to public investment in education, consistent availability over time, and prior use in studies linking school quality to later-life health and cognitive outcomes [18,20–22]. All expenditure values were inflation adjusted to 2021 dollars. The distribution of each indicator by region during the period when sample respondents were attending primary school (around age 10) are shown in S2 Figure in S1 Appendix.

We used a factor analytic approach we previously developed with these indicators from the S-ECDR dataset to derive summary measures of state education quality [22,27]. Prior to constructing the factors, we smoothed year-to-year fluctuations by calculating the three-year moving average of each indicator. We then created standardized z-scores for each of seven indicator measures by year to ensure that they were comparably scaled. The resulting factor scores represent states' relative education quality within a given 3-year period. The screeplot indicated that two factors explaining approximately 83.3% of the total variance should be retained (see S3 Figure in S1 Appendix). The first factor, "state-level education funding source," had large negative loadings of per-pupil local revenue and percent of revenue from local sources and large positive loadings of per-pupil state revenue and percent revenue from the state (see S1 Table in S1 Appendix). The second factor, "state-level education resources," had large positive loadings of per-pupil expenditures, per-pupil revenue from local sources, and teacher salary. Factors were orthogonally rotated and uncorrelated with each other.

**State Income inequality.** Income inequality is measured as the percent of income held by the top 10% of earners in the state when respondents were age 7.

**Educational attainment.** Respondents reported the number of years of schooling they completed (range: 0–17).

**Individual-level covariates.** We included sex (male or female), race/ethnicity (non-Hispanic White, non-Hispanic Black, Hispanic (any race), Other race/ethnicity), age (in years), and birth cohort (Ahead born 1913–1923, Coda born 1924–1930, HRS born 1931–1941, Warbabies born 1942–1945).

## Analytic strategy

First, we examined how demographic characteristics, education, and state-level education quality varied by U.S. region of current residence for older adults in the 2010–2018 period. Next, using multinomial logistic regression models, we estimated the relative risk of CIND and dementia prevalence in the 2010–2018 period with normal cognitive function serving as the reference category. Because some respondents had more than one observation in the 2010–2018 survey period, we report robust standard errors that account for within-person clustering over time. We show four models: a model (M1) that estimated the relative risk of dementia and CIND for current region of residence adjusting for sociodemographic characteristics of adults residing in those states, race/ethnicity, age, birth cohort and sex; a second model (M2) that adjusted for years of schooling and income inequality; a third model (M3) that additionally adjusted for state-level education quality (i.e., the state-level education funding source and state-level education resources); and a final model (M4) that included the interaction of state-level education resources with years of schooling. We do not present a model with the interaction with state-level funding source because the interaction term was not statistically significant.

To examine the percentage of the regional association that is explained by years of schooling and the two state-level education quality factors (i.e., state-level education funding source and state-level education resources), we used the Karlson–Holm–Breen (khb) method [32]. The khb method compares the coefficient for South region of residence from a model that includes education quality factors, years of schooling and respondent age, sex and race with a model controlling only for age, sex and race. We examined the amount of change in the estimated coefficient that was explained by years of schooling and the two state-level education quality factors.

Using the margins command in Stata 18 and holding all covariates at their observed values, we also calculated the predicted prevalence of dementia and CIND by region across three model specifications: 1) the base model (M1) that includes demographics (age, race and sex); 2) a model adjusting for years of schooling and demographics (M2); and 3) the final model (M4) specification that includes the above controls, the two state-level education quality factors and the interaction of state-level education resources with years of schooling.

Finally, using estimates from the final model (M4) that includes an interaction between state-level education resources and years of schooling, we calculated the predicted probability of dementia and CIND for those residing in the US South across years of schooling (8–16 years of schooling) for those who grew up in states with lower levels of education resources (resources score of −1, approximately the 20th percentile) versus higher levels of education resources (resources score of 1, approximately the 80th percentile) holding all other covariates at the sample mean. Our selected values for years of schooling reflect the expected probabilities of dementia or CIND for most levels of educational attainment (8th grade through four-years of college).

## Ethics statement

This study used secondary data and thus did not require informed consent from participants. Respondent characteristics are available from public data files, but geographic identifiers from restricted data files were used to link respondents to states. The Institutional Review Board of the University of Southern California approved this study.

## Results

### Sample characteristics

In Table 1, we summarize the characteristics of adults aged 65 and older included in our analytic sample. Across the period of observation, dementia prevalence varied by region; 9% in the South compared with 7% in the Midwest, West and Northeast. Adults living in the South were also less likely than adults living in other regions to identify as non-Hispanic White (82% in South; 93% Northeast; 92% Midwest; 85% West) and more likely to identify as non-Hispanic Black (13% in South; 6% Northeast; 6% Midwest; 4% West). Completed years of schooling also differed across regions. Adults living in the South completed the fewest years of schooling, on average, and were more likely than adults in other regions to have completed 8 or fewer years of school; 9% in the South compared with 3% in the Northeast and 4% in the Midwest and West. Adults living in the South were also more likely to have been exposed to state education systems with low education resources (−0.46), whereas adults living in the Northeast (1.10), Midwest (0.26), and West (0.68) all had exposure to state education systems that were, on average, better resourced. Individuals living in the South and West grew up in states that received more funding from state than local sources (0.47 and 0.17, respectively), in comparison to individuals living in the

**Table 1. Characteristics of U.S. born adults aged 65 and older by current region of residence, The Health and Retirement Study, 2010–2018, weighted.**

| | Northeast n = 1,219 | Midwest n = 2,350 | South n = 3,743 | West n = 1,459 |
|---|---|---|---|---|
| Dementia Classification[a] | | | | |
| Dementia | 7% | 7% | 9% | 7% |
| CIND | 20% | 19% | 20% | 18% |
| Normal | 74% | 74% | 70% | 74% |
| Education (years) | 13.1 (2.4) | 12.9 (2.4) | 12.7 (3.1) | 13.5 (2.7) |
| ≤ 8 Years of Schooling | 3% | 4% | 9% | 4% |
| 9–11 Years of Schooling | 12% | 11% | 14% | 8% |
| 12 Years of Schooling | 42% | 44% | 32% | 30% |
| 13–15 Years of Schooling | 19% | 20% | 20% | 29% |
| ≥ 16 Years of Schooling | 24% | 20% | 25% | 30% |
| State-Level Education Quality | | | | |
| Education Resources[b] | 1.1 (.9) | 0.27 (.7) | −0.46 (1.1) | 0.68 (.8) |
| Funding Source[b.] | −0.12 (.7) | −0.16 (.8) | 0.47 (.7) | 0.17 (.9) |
| Age (years) | 77.9 (7.2) | 77.7 (7.5) | 76.4 (7.1) | 77.3 (7.1) |
| Sex | | | | |
| Male | 40% | 42% | 45% | 43% |
| Female | 60% | 58% | 55% | 57% |
| Race/Ethnicity | | | | |
| White | 93% | 92% | 82% | 85% |
| Black | 6% | 6% | 13% | 4% |
| Hispanic | 1% | 1% | 3% | 10% |
| Other Race/Ethnicity | 1% | 1% | 2% | 2% |
| Person-Period Observations | 4,547 | 8,907 | 14,188 | 5,611 |

Notes:

[a]Estimates pooled across observations during 2010–2018 period. Analytic sample restricted to individuals who did not move across regions during the period.

[b]Factor summarizing state-level education quality at age 10.

Northeast and Midwest who grew up in states that received relatively more funding from local than state sources (−0.12 and −0.16, respectively).

## Multinomial logistic regression

Table 2 presents relative risk ratios (RRR) from multinomial logistic regression models predicting cognitive status (normal function is the referent group). Consistent with the descriptive analysis, adults living in the South had a greater relative risk of dementia (RRR 1.45, CI:1.22, 1.74, $p < 0.001$) or CIND (RRR 1.13, CI: 1.01, 1.27, $p < 0.05$) relative to normal cognitive function in a base model (M1) that only adjusts for the age, birth cohort, sex, and race/ethnic composition of the population. In Model 2, which additionally adjusted for years of schooling and state-income inequality, the South persisted in having higher risk of dementia (RRR 1.40, CI: 1.17, 1.68, $p < 0.001$) and CIND (RRR 1.14, CI: 1.02, 1.28, $p < .05$). However, this difference was no longer statistically significant when we included state-level funding source and state-level education resources, as shown in Model 3. Growing up in states with higher levels of education resources was associated with lower relative risk of dementia (M3: RRR 0.81, CI: 0.75, 0.87, $p < 0.001$) and CIND (M3: RRR 0.89, CI: 0.84, 0.93, $p < 0.001$) but growing up in states with education systems that relied more on state revenues than local revenues was associated with higher relative risk of dementia (M3: RRR 1.12, CI: 1.01, 1.23, $p < 0.05$) and CIND (M3: RRR 1.10, CI: 1.03, 1.16, $p < 0.01$).

## KHB method results

In Table 3, we show results from the khb method that estimated how much of the reduction in the regression coefficients for the South was explained by years of schooling and the two state-level education quality factors. Our findings suggest that state-level education resources explained 68.9% of the reduction in dementia coefficient and 59.9% of the reduction in the CIND for the South, state-level education funding sources explained about 33.3% of the reduction in the dementia coefficient and 42.6% of the CIND coefficient. Years of schooling did not explain the reduction in dementia or CIND coefficients for the South.

## Predicted dementia and CIND prevalence by region

In Fig 1, we plot predicted dementia and CIND prevalence by region using estimates from a model that adjusts for socio-demographic controls (M1), a model that further adjusts for education and state-income inequality (M2), and a model that adjusts for state-level education quality factors (M3). Consistent with descriptive and regression analyses, the predicted prevalence of dementia and CIND was higher in the South than other regions in the models that did not control for state-level education quality (M1, M2) but not significantly different than other regions after adjusting for state-level education quality factors and interactions with years of schooling (M3). Predicted values from models M1–M4 are provided in S2 Table in S1 Appendix.

## Predicted differences in probability of dementia and CIND by years of schooling and education resources

We also found evidence that the association between state-level education resources and cognitive status varied by years of schooling. We included an interaction between the two terms in Model 4. The relative risk ratios for state-level education resources for dementia (RRR 0.80, CI: 0.74, 0.87, $p < 0.001$) and CIND (RRR 0.87, CI: 0.82, 0.91, $p < 0.001$) as well as the relative risk ratios for years of schooling for dementia (RRR 0.73, CI: 0.70, 0.75, $p < 0.00$) and CIND (RRR 0.80, CI: 0.78, 0.81, $p < 0.001$) show that state-level education resources and years of schooling were associated with lower relative risk of dementia and CIND. However, the interaction term was statistically significant for both dementia (RRR 1.05, CI: 1.02, 1.08, $p < 0.001$) and CIND (RRR 1.03, CI: 1.01, 1.04, $p < 0.01$), suggesting that associations between state-level education resources and cognitive status were attenuated among those with more years of schooling.

**Table 2. Relative risk ratios from multinomial regressions predicting cognitive status among adults 65 and older, Health and Retirement Study 2010-2018.**

| | M1 RRR (95% CI) | M2 RRR (95% CI) | M3 RRR (95% CI) | M4 RRR (95% CI) |
|---|---|---|---|---|
| **Dementia (Ref = Normal)** | | | | |
| Region at Interview (Ref = Midwest) | | | | |
| Northeast | 0.92 | 0.99 | 1.18 | 1.18 |
| | (0.73, 1.15) | (0.78, 1.26) | (0.92, 1.51) | (0.92, 1.51) |
| South | 1.45*** | 1.40*** | 1.13 | 1.09 |
| | (1.22, 1.74) | (1.17, 1.68) | (0.93, 1.37) | (0.90, 1.33) |
| West | 0.96 | 1.27* | 1.34* | 1.31* |
| | (0.76, 1.20) | (1.00, 1.62) | (1.05, 1.72) | (1.03, 1.69) |
| Years of Schooling | | 0.71*** | 0.72*** | 0.73*** |
| | | (0.69, 0.73) | (0.70, 0.74) | (0.70, 0.75) |
| State-Level Income Inequality | | 1.00 | 1.00 | 1.00 |
| | | (0.98, 1.01) | (0.99, 1.02) | (0.99, 1.02) |
| State-Level Funding Source | | | 1.12* | 1.11* |
| | | | (1.01, 1.23) | (1.01, 1.22) |
| State-Level Education Resources | | | 0.81*** | 0.80*** |
| | | | (0.75, 0.87) | (0.74, 0.87) |
| Years of Schooling x Education Resources | | | | 1.05*** |
| | | | | (1.02, 1.08) |
| **CIND (Ref = Normal)** | | | | |
| Region at Interview (Ref = Midwest) | | | | |
| Northeast | 1.01 | 1.05 | 1.16+ | 1.17* |
| | (0.87, 1.16) | (0.91, 1.22) | (1.00, 1.35) | (1.00, 1.35) |
| South | 1.13* | 1.14* | 0.99 | 0.98 |
| | (1.01, 1.27) | (1.02, 1.28) | (0.88, 1.12) | (0.87, 1.11) |
| West | 0.91 | 1.12 | 1.15+ | 1.14+ |
| | (0.80, 1.05) | (0.98, 1.29) | (0.99, 1.32) | (0.99, 1.31) |
| Years of Schooling | | 0.79*** | 0.80*** | 0.80*** |
| | | (0.78, 0.81) | (0.78, 0.81) | (0.78, 0.81) |
| State-Level Income Inequality | | 1.00 | 1.01 | 1.01 |
| | | (0.99, 1.01) | (1.00, 1.02) | (1.00, 1.02) |
| State-Level Funding Source | | | 1.10** | 1.09** |
| | | | (1.03, 1.16) | (1.03, 1.16) |
| State-Level Education Resources | | | 0.89*** | 0.87*** |
| | | | (0.84, 0.93) | (0.82, 0.91) |
| Years of Schooling x Education Resources | | | | 1.03** |
| | | | | (1.01, 1.04) |

Note: M1-M4 control for age, gender, and race/ethnicity. N = 8,771 respondents contributing 33,253 observations. Standard errors clustered by respondent. Education centered at 12 years of schooling. Higher values on state-level education resources and state-level funding source factors reflect greater state-level resources for public schools and greater state funding of public schools, respectively.

+ p < .10, * p < .05, **p < .01 ***p < .001.

**Table 3. Estimates from KHB analysis decomposing changes in regression coefficients for U.S. South region of current residence by education and state-level education quality factors.**

| | Dementia | CIND |
|---|---|---|
| | b | b |
| Reduced South Region (ref = Midwest)[a] | 0.34*** | 0.13* |
| Full South Region[b] | 0.13 | −0.00 |
| $b_{reduced} - b_{full}$ | 0.21*** | 0.13*** |
| **% of South coefficient explained by:** | | |
| Years of Schooling | −2.3 | −2.4 |
| State-Level Funding Source | 33.3*** | 42.6*** |
| State-Level Education Resources | 68.9*** | 59.9*** |

Notes:

[a]Estimates from model adjusting for age, race, and gender.

[b]Estimates from model adjusting for age, race, gender, years of schooling, and state-level education quality indicators.

+ p < .10, * p < .05, **p < .01 ***p < .001.

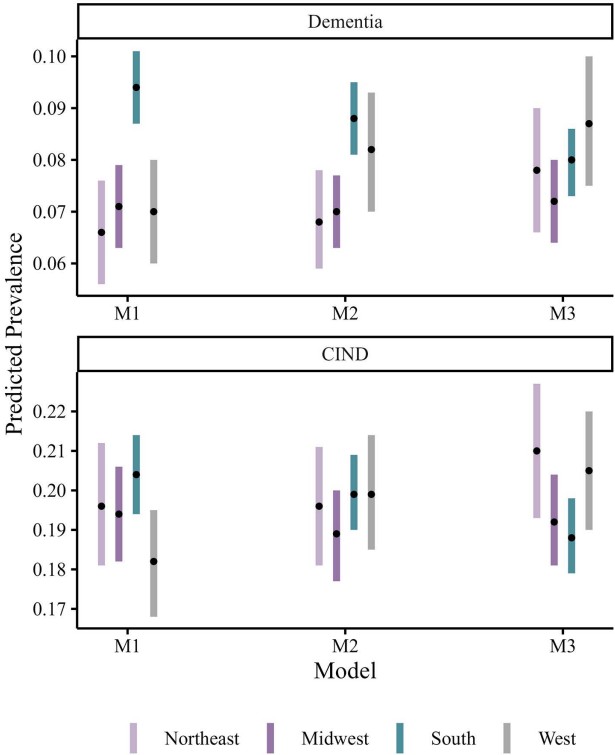

**Figure 1. Predicted probability of Dementia and CIND for individual with average characteristics by region.** Notes: Margins command used to calculate probability of dementia varying region of residence but holding all other covariates at observed values. Points represent predicted values; lines represent 95% confidence interval around estimates. M1: Model 1 estimates adjusted for age, race/ethnicity, and gender; M2: Model 2 estimates adjusted for age, race/ethnicity, gender, and years of schooling; M4: Model 4 estimates adjusted for age, race/ethnicity, gender, years of schooling, state-level education funding source, state-level education resources, state-level education resources and years of schooling interaction.

For ease of interpretation, in Fig 2 we show the predicted probability of dementia and CIND by years of schooling and state-level education resources for individuals living in the South. These predicted probabilities demonstrate that state-level education resources were more strongly associated with cognitive status among those with low levels of education. For individuals with 8 years of schooling, the predicted probability of dementia was 0.19 (CI: 0.16, 0.21) among those who grew up in states with low-resourced education systems compared to 0.11 (CI: 0.09, 0.11) for those who grew up in states with highly resourced education systems. Among persons with 12 years of schooling, the predicted probability of dementia was 0.07 (CI: 0.06, 0.08) for those who grew up in states with low-resourced education systems compared to 0.05 (CI: 0.04, 0.06) among those who grew up in states with highly resourced education systems. For those who completed 16 years of schooling, the predicted probability of dementia was 0.02 (CI: 0.01, 0.02) regardless of the level of resources of the education system in the state where they resided as children. Thus, the resource level of a state education system was strongly associated with dementia risk for those with low levels of educational attainment but not among those with high levels of educational attainment (see S3 Table in S1 Appendix for predicted differences and confidence intervals by

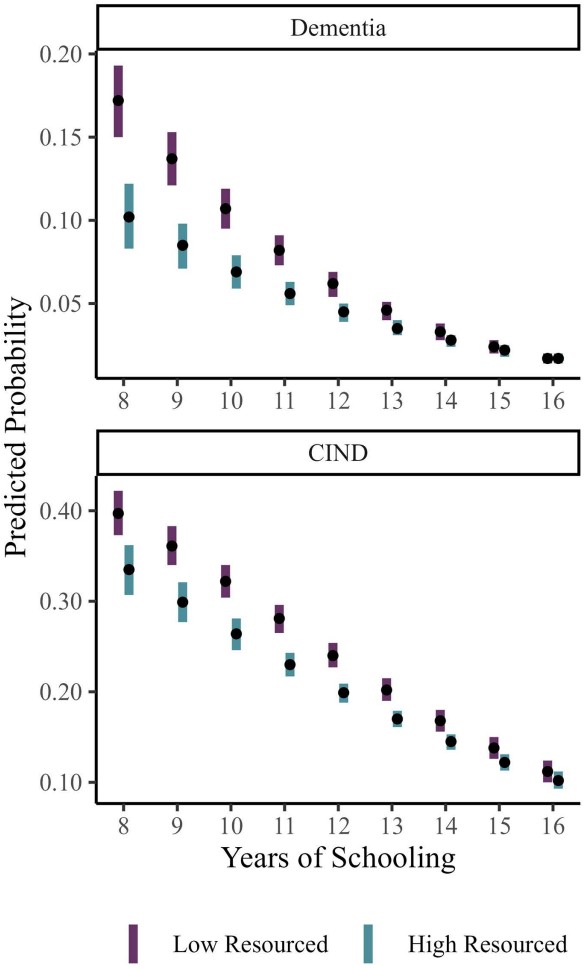

**Figure 2. Predicted probability of Dementia and CIND in U.S. South by years of schooling and state-level education resources.** Margins command used to calculate predicted probability of dementia or CIND at 8 to 16 years of schooling varying state-level education resources from low (−1) to high (1) and setting region of residence to be South. Points represent predicted values; lines represent 95% confidence interval around estimates. All other covariates are held constant at their mean. Predicted probabilities calculated from M4 estimates.

state-level education resources across years of schooling). As shown in Fig 2, a similar pattern was also observed for CIND. This interaction is important for understanding the Southern disadvantage in dementia and CIND as a much larger share of older adults with low education were living in the South and those with low education seem to be particularly at risk of dementia and CIND when they grew up in states with low-resourced education systems.

## Sensitivity analyses

We estimated several alternative specifications to test the robustness of our reported findings. First, we estimated region stratified multinomial logistic models that allowed the association of covariates with cognitive status to vary across region. The associations between state-level education quality factors and cognitive status were strongest in the South, though we observed a similar pattern of results in the Northeast and Midwest region as well (see S4–S7 Tables in S1 Appendix). We also re-estimated the khb analysis after stratifying by education level (12 or fewer years of schooling versus 13 or more years of schooling). We found that among those with 12 or fewer years of schooling there is a statistically significant association between South region of residence and cognitive status (see S8 Table in S1 Appendix) in the unadjusted model, and this association becomes non-significant with the inclusion of years of schooling and state-level education quality factors (Dementia $b_{reduced} - b_{full}$ 0.47; CIND $b_{reduced} - b_{full}$ 0.31). For those with 13 or more years of schooling, South region of residence is not associated with cognitive status and the coefficient for South region of residence is reduced to a lesser degree after including years of schooling and education quality factors (Dementia $b_{reduced} - b_{full}$ 0.10; CIND $b_{reduced} - b_{full}$ 0.05). Finally, we restricted our sample to respondents who reported living in the same region at birth, at age 10, and at the time of the 2010 HRS interview. Results from this sensitivity analysis (see S9 Table in S1 Appendix) were consistent with the main analysis; people who lived in the South had a higher prevalence of dementia (RRR Dementia 1.75, CI: 1.39, 2.20, $p < 0.001$) and CIND (RRR 1.35, CI: 1.18, 1.55, $p < 0.001$) in a model adjusting for demographic characteristics, years of schooling, and income inequality, but this difference was no longer statistically significant after accounting for education quality.

## Discussion

Regional disparities in cognitive impairment and dementia are well-documented, but the drivers of these differences are poorly understood. This study is among the first to explore the extent to which educational quality, in addition to attainment, contributes to regional disparities in CIND and dementia. The study findings provide evidence that in addition to attainment, state-level education quality is also an important determinant of cognitive impairment and dementia, that it explains more of the observed geographic disparities in cognitive health than educational attainment, and that higher state education resources were particularly beneficial at lower levels of educational attainment.

We found, as others have previously, that the risk of dementia and CIND is significantly higher in the U.S. South and that some of this geographic disparity is explained by accounting for differences in educational attainment [5–9]. However, when we considered both educational attainment and quality, we found that educational attainment alone did not account for the elevated prevalance of dementia and CIND in the South. In contrast, education quality fully explained the regional disparities.

These findings suggest differences in education quality – not just attainment – have played a central role in shaping the South's relative disadvantage in cognitive aging. Higher quality education may provide greater access to cognitively enriching activities during childhood that benefit cognitive aging in later life [21,33,34], and this critical early cognitive development may be better captured with school quality than educational attainment [17,35], particularly in the South where a much larger portion of older adults had 8 years of schooling or less (9% in the South versus 3–4% in the other regions). The historical underinvestment in public schools in the South, relative to other regions, very likely limited access to key early life cognitively enriching opportunities that would protect against cognitive impairment and dementia risk in older ages.

Our finding that state-level education resources is associated with CIND and dementia risk is consistent with prior theory and empirical evidence suggesting there are cognitive developmental and health benefits of richer school resource environments [18–24]. We also found that older adults who attended schools in states with a greater share of education funding from state, rather than local, sources had a higher risk of CIND and dementia. In the early to mid-20th century, this pattern was especially characteristic of Southern states, where limited local tax bases and broader economic underdevelopment led to a stronger role for state governments in financing public schools [36–38]. However, state-level control in the South often served to entrench educational inequalities, rather than correct them. Many state governments maintained racially and economically stratified systems by prioritizing vocational training and limiting investment in academic curricula [36] —decisions that may have constrained cognitive development and shaped long-term occupational trajectories in ways that increase later-life dementia risk [39]. In contrast, states like California also expanded the state share of education funding during this period, but did so primarily to stabilize school systems amid fiscal crisis and rapid demographic change during and after the Great Depression [38]. These historical differences in how and why states relied on state-level funding may help contextualize our finding that regional disparities in dementia were fully explained after accounting for education resources and funding structures—particularly in the South, but not in the West.

We found an interaction between years of schooling and educational resources that indicated that exposure to higher education quality is particularly advantageous for those with less education. Similar findings have been reported previously. One study of community-dwelling adults ages 65 and older in Alabama reported an association between county-level school term length and cognitive function, but only among older adults with less than high school educational attainment [18]. Another study of older community-dwelling adults in Wisconsin found a stronger association between higher school quality and later life cognition among those with lower academic achievement in high school [19]. This differential impact of school quality by educational attainment is consistent with research showing that less advantaged children benefit more from higher quality education compared to more advantaged children [40]. Higher quality education may enhance development of both cognitive and noncognitive skills that can later be leveraged to secure better life course opportunities, such as more cognitively stimulating jobs, independent of level of education. Moreover, our finding that exposure to higher education quality is particularly advantageous for the least educated aligns with prior research on other health outcomes. Others, for instance, have found that higher educational attainment acts as a protective factor, reducing the adverse effects of unfavorable state contexts on disability rates, and that the least educated were the most susceptible to the negative impacts of state policies and environments [41]. Thus, the role of education quality in later life cognitive health may be especially important among the most disadvantaged groups.

We also found a higher prevalence of dementia in the West, relative to the Midwest, only after adjusting for educational attainment. This could reflect selection into Western states on health and socioeconomic factors, such as education. In this study, the proportion of older adults with higher education was greatest in the West compared to other regions; there was an especially large gap in college education levels between these regions. Others have reported a high level of regional mobility in the HRS cohorts – one-third of respondents reside in a different region at the time of the interview from the region where they were born – with greater mobility into the West and among the college-educated [9]. Thus, the overall lower dementia prevalence observed in the West may largely be driven by the high levels of schooling among the older adults who live there, an advantage that disappears when we account for differential education levels across regions.

Although this study provides novel insights into the role of education quality on geographic disparities in CIND and dementia risk, several limitations should be noted. Our education quality measures reflect average conditions at the state level and cannot account for local variation in school environments. While this is a limitation, state governments played a central role in shaping public education during the first half of the 20th century – particularly in the South, where weaker local tax bases, delayed implementation of property taxes, and slower economic development led to heavier reliance on state-level funding for schools. Given the absence of consistent district-level data for cohorts born between 1913 and

1945, state-level indicators offer the most historically grounded and policy-relevant measures of early-life educational conditions.

We also made necessary but strong assumptions when assigning respondents to the state in which they were educated. We used the best available information – state of residence during school or at approximately age 10 – but cannot confirm whether this reflects where respondents lived throughout all years of schooling. We do not know, however, if this was the state of residence during all years of schooling. In supplemental analyses, we restricted the sample to respondents who lived in the same region at birth, during school, and at the time of their interview and found a similar, if not somewhat stronger pattern of findings.

It is possible that selective survival may bias our estimates. Individuals who were exposed to lower-quality education or other early-life disadvantages may have been less likely to survive into the period of HRS observation, which could attenuate observed associations between education quality and cognitive outcomes. In addition, our focus on birth cohorts between 1913 and 1945 allows us to examine differences in early educational conditions, but it does not fully disentangle cohort effects from broader period-specific influences. Although we standardized education quality indicators by year to adjust for secular trends in school investment, our design cannot rule out the influence of unmeasured period effects that may vary across regions or states. We also acknowledge that some indicators of education quality, such as teacher salary or per-pupil spending, may partially reflect broader state-level economic conditions; although we were unable to adjust for factors such as state GDP or median wages, our findings remained robust after adjusting for state-level income inequality, suggesting that the observed associations are not solely driven by economic disparities.

Finally, this study focused specifically on understanding early-life education environments as a key explanatory factor for regional disparities in cognitive impairment and dementia. Future research is needed that explores how educational quality interacts with regionally patterned health and behavioral risk factors that may also contribute to dementia risk – such as cardiovascular and cerebrovascular health, smoking and obesity – in order to better clarify the causal pathways linking early life education quality to regional disparities in cognitive aging.

It has become abundantly clear that the U.S. South fares worse on many health indicators, including cognitive impairment and dementia. Our study highlights the importance of education quality in understanding regional disparities in cognitive impairment and dementia. Prior research using the HRS has reported that region of birth is more strongly associated with dementia prevalence than current region of residence and attributed this to the influence of early-life place-based exposures [9]. Our findings suggest that early educational conditions – particularly in under-resourced Southern states – play a central role in shaping later-life cognitive health. The cohorts in our study, born between 1913 and 1945, attended school during a period when state investment in public education was beginning to expand, but before the implementation of federal reforms in the 1960s. Legislation such as the Civil Rights Act of 1964 and the Elementary and Secondary Education Act of 1965 introduced major changes in school funding, desegregation enforcement, and instructional support – but these efforts came too late to substantially affect the educational experiences of most individuals in our sample. Because our measure of education quality is anchored at age 10, our findings likely underestimate the long-term cognitive benefits of later policy interventions. Future research should examine the joint importance of educational attainment and quality across different schooling stages (i.e., pre-K, primary, secondary, and college) on cognitive impairment and dementia as new cohorts with very different educational experiences reach older ages.

## Supporting information

**S1 Appendix. Supplemental Figures and Tables on Education Quality and CIND/Dementia Status in Older U.S. Adults.** This appendix includes additional figures and tables referenced in the manuscript, including descriptive analyses of state-level education quality, predicted probabilities of dementia and CIND, region-specific regression models, and KHB decomposition results.
(DOCX)

## Author contributions

**Conceptualization:** Jennifer A. Ailshire, Katrina M. Walsemann.

**Data curation:** Katrina M. Walsemann.

**Formal analysis:** Heide Jackson.

**Funding acquisition:** Katrina M. Walsemann.

**Investigation:** Jennifer A. Ailshire.

**Methodology:** Jennifer A. Ailshire, Katrina M. Walsemann.

**Supervision:** Jennifer A. Ailshire, Katrina M. Walsemann.

**Visualization:** Heide Jackson.

**Writing – original draft:** Jennifer A. Ailshire, Heide Jackson.

**Writing – review & editing:** Jennifer A. Ailshire, Katrina M. Walsemann, Mateo P. Farina, Heide Jackson.

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
