## [Decision Letter · Decision Letter 0]

3 Feb 2025

PONE-D-24-28756The role of educational attainment and quality in U.S. regional variation in prevalence of dementia and CINDPLOS ONE

Dear Dr. Ailshire,

Thank you for submitting your manuscript to PLOS ONE. After careful consideration, we feel that it has merit but does not fully meet PLOS ONE’s publication criteria as it currently stands. Therefore, we invite you to submit a revised version of the manuscript that addresses the points raised during the review process.

We look forward to receiving your revised manuscript.

Kind regards,

Ryan G Wagner, MSc(Med), MBBCh, PhD

Academic Editor

PLOS ONE

“This work was funded by the National Institute on Aging (grants R01AG055481 and P30AG043073).”

Additional Editor Comments:

I commend the authors on a very well written and interesting manuscript. In addition to the detailed and very useful comments provided by the Reviewers, I would ask the authors to consider the following:

1.) Kindly clarify whether the HRS collects information on individuals' 'gender' or 'sex' and, if necessary, update the manuscript accordingly. 

2.) The authors do a good job in describing the various measures and approach that they used to assess education quality. Justifying the use of these seven measures (either by citing previous literature or acknowledging novelty in their approach) might further strengthen the manuscript. 

3.) Finally, I wonder whether determinants for cognitive impairment can truly be explored in isolation. To this point, we know that the southern US represents the 'stroke belt' and is home to some of the highest levels of cardio-/cerebrovascular risk in the US. To what effect might this have on the results derived in this study and could some of the variation in prevalence by region be explained by clinical/biological determinants that vary by region (and may very well be related to education or socioeconomic status)? Is there a way for the authors to adjust for some of these potential biological risk factors- or explore the potential interaction between these in the future? 

Reviewers' comments:

Reviewer's Responses to Questions

**Comments to the Author**

1. Is the manuscript technically sound, and do the data support the conclusions?

Reviewer #1: Partly

Reviewer #2: Yes

2. Has the statistical analysis been performed appropriately and rigorously? 

Reviewer #1: Yes

Reviewer #2: Yes

3. Have the authors made all data underlying the findings in their manuscript fully available?

Reviewer #1: No

Reviewer #2: No

4. Is the manuscript presented in an intelligible fashion and written in standard English?

Reviewer #1: Yes

Reviewer #2: Yes

5. Review Comments to the Author

Reviewer #1: This is a well executed paper on an important topic.

My only concern is regarding the measure of "educational quality," which may be acting as a proxy for state wealth or income. Numerous studies have demonstrated a socioeconomic gradient in dementia, where factors like income and wealth, alongside education, play a significant role. For example, a higher "teaching salary" in one state may simply indicate a higher average wage in that region, rather than an objective improvement in educational quality.

The authors should consider doing more to differentiate between the impact of "educational quality" and that of wealth or income, both of which can exert independent effects on dementia risk. For instance, wealthier states might offer more high-paying jobs, better amenities, or enhanced healthcare infrastructure, all of which could influence the outcomes observed, irrespective of educational quality.

I think the authors could account for the above by adjusting for state/regional price parities in the education quality measures or, at the very least, controlling for variables such as state GDP, median wages, or similar economic indicators. This would help to ensure that the findings truly reflect differences in educational quality, rather than economic disparities.

Reviewer #2: Thank you for the opportunity to review this fascinating manuscript on educational quality and dementia disparities. The manuscript is well-written, and I only have a few comments below.

1. While the paper makes an important contribution by highlighting education quality’s role in regional dementia disparities, it could better situate these findings within broader discussions of policy implications for reducing geographic health disparities, the changing landscape of educational inequality, and intersections with other social determinants of cognitive aging. These contextual discussions would enhance the paper’s relevance and impact for research and policy audiences.

2. The limitations section could more explicitly address potential bias arising from using state-level rather than district-level education quality measures, issues related to selective mortality in the study population, and challenges in distinguishing between cohort and period effects in the analysis. These methodological considerations would provide readers with a more complete understanding of the study’s constraints and interpretation of findings.

3. The figures would benefit from explicit documentation of the uncertainty estimates in the visualizations. While both figures display point estimates (represented by dots) and what appear to be 95% confidence intervals (shown as vertical bars, right?), these elements are not explicitly identified in the figure legends. For proper statistical reporting and interpretation, I recommend revising the figure captions to include:

For Figure 1: “… Point estimates (dots) and 95% confidence intervals (vertical bars) are shown for each region and model …”

Similar to Figure 2.

4. Please provide citations for cognitive reserve theory in Introduction.

5. The authors may need to explicitly state the number of cases that were removed due to missing.

6. Why did the authors use HRS only up to 2016 despite HRS having more recent waves?

7. Please spell out the abbreviation (e.g., RRR) the first time it appears.

6. PLOS authors have the option to publish the peer review history of their article (what does this mean? ). If published, this will include your full peer review and any attached files.

**Do you want your identity to be public for this peer review?** For information about this choice, including consent withdrawal, please see our Privacy Policy .

Reviewer #1: No

Reviewer #2: No

---

## [Author Response · Author response to Decision Letter 1]

2 Aug 2025

1. We have reviewed all stylistic aspects of the manuscript to ensure that it aligns with PLOS ONE requirements.

2. We have included an amended Role of Funder statement in our cover letter, as requested.

3. We provide here our revised statement on data availability.

All data underlying the findings described in this manuscript are available, though one dataset has some restrictions. The State-Education Contextual Data Resource (SECD-R) is publicly available and can be downloaded from OpenICPSR (https://doi.org/10.3886/E233063V1). State-level data on income inequality are available from the Economic Policy Institute and can be downloaded directly here: http://go.epi.org/unequalstates2018data.Individual-level survey data used in this manuscript are publicly available from the Health and Retirement Study (HRS) and can be accessed from their website (https://hrsdata.isr.umich.edu/data-products). Data used to identify respondent state of residence during schooling years contain potentially identifying information and are thus provided as restricted data products by HRS through a secure data enclave. Users can apply for access to these data through the HRS website (https://hrs.isr.umich.edu/data-products/restricted-data/vdi).

4. We have included captions for Supporting Information files at the end of the manuscript

5. We have reviewed the reference list to ensure that it is complete and correct. None of the papers we cited have been listed as retracted. All references that have been added in this revision are listed below in response to reviewer comments.

Additional Editor Comments:

1) Kindly clarify whether the HRS collects information on individuals' 'gender' or 'sex' and, if necessary, update the manuscript accordingly.

Response: Respondent is first recorded through a household roster during sample selection that requires either self-identification, or identification by a household member, that the person is male or female. In follow-up waves interviewers are asked to confirm the sex of the respondent. So technically HRS collects information about sex. However, in HRS data production they call this variable ‘gender’ and we followed that in the paper. However, we understand the preference for fidelity to the survey wording itself and so we have replaced any instance of ‘gender’ in the paper with ‘sex’.

2.) The authors do a good job in describing the various measures and approach that they used to assess education quality. Justifying the use of these seven measures (either by citing previous literature or acknowledging novelty in their approach) might further strengthen the manuscript.

Response: We have added additional information to the description of the measures of education quality in the Methods section, including citing several studies that have used similar measures and a particular study that used this specific approach.

References:

18. Crowe M, Clay OJ, Martin RC, Howard VJ, Wadley VG, Sawyer P, et al. Indicators of Childhood Quality of Education in Relation to Cognitive Function in Older Adulthood. The Journals of Gerontology: Series A. 2013;68: 198–204. doi:10.1093/gerona/gls122

20. Sisco S, Gross AL, Shih RA, Sachs BC, Glymour MM, Bangen KJ, et al. The Role of Early-Life Educational Quality and Literacy in Explaining Racial Disparities in Cognition in Late Life. J Gerontol B Psychol Sci Soc Sci. 2015;70: 557–567. doi:10.1093/geronb/gbt133

21. Soh Y, Whitmer RA, Mayeda ER, Glymour MM, Peterson RL, Eng CW, et al. State-Level Indicators of Childhood Educational Quality and Incident Dementia in Older Black and White Adults. JAMA Neurol. 2023 [cited 7 Apr 2023]. doi:10.1001/jamaneurol.2022.5337

22. Walsemann KM, Jackson H, Abbruzzi E, Ailshire JA. State-level education quality and trajectories of cognitive function by race and educational attainment. The Milbank Quarterly 2024;102. https://doi.org/10.1111/1468-0009.12709.

27. Walsemann, K.M., Abbruzzi, E., Jackson, H.M. et al. Historical state-level data on U.S. public school systems from 1919/20 to 1973/74. Sci Data 12, 1218 (2025). doi.org/10.1038/s41597-025-05507-6

3.) Finally, I wonder whether determinants for cognitive impairment can truly be explored in isolation. To this point, we know that the southern US represents the 'stroke belt' and is home to some of the highest levels of cardio-/cerebrovascular risk in the US. To what effect might this have on the results derived in this study and could some of the variation in prevalence by region be explained by clinical/biological determinants that vary by region (and may very well be related to education or socioeconomic status)? Is there a way for the authors to adjust for some of these potential biological risk factors- or explore the potential interaction between these in the future?

Response: We appreciate this observation and agree that regional differences in these risks may contribute to geographic variation in cognitive outcomes, and these risks are likely shaped in part by early-life socioeconomic and educational conditions. In this study, our aim was specifically to determine the contribution of state-level education quality to regional disparities in dementia and cognitive impairment, above and beyond educational attainment, so we didn’t focus on all the potential factors that may lie on the causal pathway linking early education quality to later-life cognitive impairment and dementia. However, we agree that future work would benefit from explicitly modeling these potential pathways and we now note in the conclusion that future studies should investigate how health and behavioral risk factors interact with early-life education environments to shape cognitive aging across regions of the U.S.

Reviewers' comments:

Reviewer #1: This is a well executed paper on an important topic.

My only concern is regarding the measure of "educational quality," which may be acting as a proxy for state wealth or income. Numerous studies have demonstrated a socioeconomic gradient in dementia, where factors like income and wealth, alongside education, play a significant role. For example, a higher "teaching salary" in one state may simply indicate a higher average wage in that region, rather than an objective improvement in educational quality.

The authors should consider doing more to differentiate between the impact of "educational quality" and that of wealth or income, both of which can exert independent effects on dementia risk. For instance, wealthier states might offer more high-paying jobs, better amenities, or enhanced healthcare infrastructure, all of which could influence the outcomes observed, irrespective of educational quality.

I think the authors could account for the above by adjusting for state/regional price parities in the education quality measures or, at the very least, controlling for variables such as state GDP, median wages, or similar economic indicators. This would help to ensure that the findings truly reflect differences in educational quality, rather than economic disparities.

Response: We appreciate the reviewer’s concern and agree that differentiating educational quality from broader state economic conditions is important. To address this, we included state-level income inequality at age 7 as a covariate in all models. This variable—calculated as the share of income held by the top 10% versus the bottom 90% based on IRS tax records—captures structural variation in economic stratification across states during early childhood and has been used in prior work to account for economic context (Sommeiller & Price, 2018; Walsemann et al., 2024). Income inequality was not significantly associated with cognitive status (normal cognition, cognitive impairment without dementia, or dementia) and did not alter the associations between state-level education quality and cognitive outcomes. This suggests that our education quality factor is not merely capturing overall state affluence or income distribution—particularly given that our indicators were standardized by year, allowing us to model relative differences in educational investment within cohort-specific economic contexts. We now clarify this in the manuscript. We have added a description of this additional variable to the Methods section, and include in the tables and text results. We also acknowledge in the Discussion that while our findings appear robust to adjustment for state income inequality, we weren’t able to account for additional indicators of state economic context.

Reviewer #2: Thank you for the opportunity to review this fascinating manuscript on educational quality and dementia disparities. The manuscript is well-written, and I only have a few comments below.

1. While the paper makes an important contribution by highlighting education quality’s role in regional dementia disparities, it could better situate these findings within broader discussions of policy implications for reducing geographic health disparities, the changing landscape of educational inequality, and intersections with other social determinants of cognitive aging. These contextual discussions would enhance the paper’s relevance and impact for research and policy audiences.

Response: We revised the discussion section to better situate our findings within the broader historical and policy context. The cohorts in our study, born between 1913 and 1945, experienced schooling during a period of gradual—but uneven—investment in public education. Many states increased funding for public schools, expanded access, and began to improve teacher salaries and instructional resources. These efforts laid the groundwork for more substantial reforms in the decades that followed. The most significant and coordinated reforms—including the Civil Rights Act of 1964 and the Elementary and Secondary Education Act of 1965—occurred after most individuals in our sample had completed their primary education. Because our education quality measure captures conditions around age 10, it reflects school environments that, while beginning to improve, still predated these major policy shifts. As a result, our estimates likely understate the longer-term impact of exposure to higher-quality schooling, particularly for students in historically underfunded and segregated systems. We now reflect this in the revised discussion.

2. The limitations section could more explicitly address potential bias arising from using state-level rather than district-level education quality measures, issues related to selective mortality in the study population, and challenges in distinguishing between cohort and period effects in the analysis. These methodological considerations would provide readers with a more complete understanding of the study’s constraints and interpretation of findings.

Response: We have revised the limitations section to more explicitly address these important methodological considerations.

First, we now clarify the rationale for using state-level measures of education quality. While we recognize that education is administered locally, state governments have historically exerted substantial control over school funding formulas, resource distribution, and curricular standards—particularly during the mid-20th century. State-level data capture meaningful variation in these structural conditions. Longitudinal, comparable district-level data are not available for older cohorts, particularly for the years and states covered in our study. As a result, state-level indicators remain the most consistent and historically grounded approach for capturing exposure to educational conditions during early life. Future research using more recent cohorts may be able to incorporate finer-grained local data to examine within-state disparities in school quality and their implications for cognitive outcomes.

Second, we now acknowledge that selective mortality may bias our results. Individuals who were exposed to more disadvantaged early-life conditions may have been less likely to survive into the HRS observation window, potentially attenuating associations between early education quality and later-life cognitive outcomes. This limitation is common to studies using older adult samples, and while selective survival may reduce the observed magnitude of these associations, it is unlikely to fully account for the patterns we observe. We now explicitly acknowledge this possibility in the limitations section.

Finally, we acknowledge the analytic challenge of distinguishing cohort effects from period effects. Our study was designed to assess the long-term impact of early-life educational conditions by focusing on individuals born between 1913 and 1945 and linking them to state-level education quality measures centered at age 10. This approach captures historical differences in schooling conditions across birth cohorts, rather than variation by calendar year. While standardizing indicators by year helps account for national shifts in school resources, it does not fully isolate cohort from period effects. We now clarify this limitation in the manuscript.

3. The figures would benefit from explicit documentation of the uncertainty estimates in the visualizations. While both figures display point estimates (represented by dots) and what appear to be 95% confidence intervals (shown as vertical bars, right?), these elements are not explicitly identified in the figure legends. For proper statistical reporting and interpretation, I recommend revising the figure captions to include:

For Figure 1: “… Point estimates (dots) and 95% confidence intervals (vertical bars) are shown for each region and model …”

Similar to Figure 2.

Response: Thank you for this suggestion, we now include the following in the figure caption for Figures 1 and 2: “ Points represent predicted values; lines represent 95% confidence interval around estimates. “

4. Please provide citations for cognitive reserve theory in Introduction.

Response: We have added two citations on cognitive reserve to the Introduction.

13. Barulli D, Stern Y. Efficiency, capacity, compensation, maintenance, plasticity: emerging concepts in cognitive reserve. Trends Cogn Sci. 2013;17(10):502-509. doi.org/10.1016/j.tics.2013.08.012

14. Mungas D, Gavett B, Fletcher E, Farias ST, DeCarli C, Reed B. Education amplifies brain atrophy effect on cognitive decline: implications for cognitive reserve. Neurobiol Aging. 2018;68:142-150. doi.org/10.1016/j.neurobiolaging.2018.04.002

5. The authors may need to explicitly state the number of cases that were removed due to missing.

Response: We have added a flow chart showing this information in S1 Figure.

6. Why did the authors use HRS only up to 2016 despite HRS having more recent waves?

Response: We have updated the HRS data we use in our analysis to 2018. We chose not to use the 2020 wave given challenges in data collection during COVID. Early release 2022 data are now available, but we prefer to use only HRS final data products as we have found errors in early release data in the past. The text and tables have been updated to reflect the additional wave of data.

7. Please spell out the abbreviation (e.g., RRR) the first time it appears.

Response: In the revised manuscript RRR is spelled out the first time it appears.

---

## [Decision Letter · Decision Letter 1]

31 Aug 2025

The role of educational attainment and quality in U.S. regional variation in prevalence of dementia and CIND

PONE-D-24-28756R1

Dear Dr. Ailshire,

We’re pleased to inform you that your manuscript has been judged scientifically suitable for publication and will be formally accepted for publication once it meets all outstanding technical requirements.

Kind regards,

Ryan G Wagner, MSc(Med), MBBCh, PhD

Academic Editor

PLOS ONE

Reviewer's Responses to Questions

**Comments to the Author**

1. If the authors have adequately addressed your comments raised in a previous round of review and you feel that this manuscript is now acceptable for publication, you may indicate that here to bypass the “Comments to the Author” section, enter your conflict of interest statement in the “Confidential to Editor” section, and submit your "Accept" recommendation.

Reviewer #1: All comments have been addressed

Reviewer #2: All comments have been addressed

2. Is the manuscript technically sound, and do the data support the conclusions?

Reviewer #1: Yes

Reviewer #2: Yes

3. Has the statistical analysis been performed appropriately and rigorously? 

Reviewer #1: Yes

Reviewer #2: Yes

4. Have the authors made all data underlying the findings in their manuscript fully available?

Reviewer #1: Yes

Reviewer #2: No

5. Is the manuscript presented in an intelligible fashion and written in standard English?

Reviewer #1: Yes

Reviewer #2: Yes

6. Review Comments to the Author

Reviewer #1: The authors have done a satifactory job addressing all of the comments in the revised manuscript.

A punctuation mark (.) is missing in the abstract before "These findings suggest".

Reviewer #2: The authors have addressed all my comments, and I have no further comments. It is my pleasure to review this paper.

7. PLOS authors have the option to publish the peer review history of their article (what does this mean? ). If published, this will include your full peer review and any attached files.

**Do you want your identity to be public for this peer review?** For information about this choice, including consent withdrawal, please see our Privacy Policy .

Reviewer #1: No

Reviewer #2: No

---

## [Editor Report · Acceptance letter]

PONE-D-24-28756R1

PLOS ONE

Dear Dr. Ailshire,

I'm pleased to inform you that your manuscript has been deemed suitable for publication in PLOS ONE. Congratulations! Your manuscript is now being handed over to our production team.

Kind regards,

on behalf of

Prof. Ryan G Wagner

Academic Editor

PLOS ONE